# Analysis of Chemical Constituents of Traditional Chinese Medicine Jianqu before and after Fermentation Based on LC-MS/MS

**DOI:** 10.3390/molecules28010053

**Published:** 2022-12-21

**Authors:** Yishuo Wang, Ruisheng Wang, Zhenling Zhang, Yitian Chen, Mengmei Sun, Jia Qiao, Ziwei Du

**Affiliations:** 1College of Pharmacy, Henan University of Chinese Medicine, Zhengzhou 450046, China; 2Henan Integrated Engineering Technology Research Center of Traditional Chinese Medicine Production, Zhengzhou 450046, China

**Keywords:** UPLC-QTOF-MS/MS technology, jianqu, fermentation, chromatographic peak identification, chemical composition

## Abstract

Objective: To detect the chemical constituents in Jianqu samples under different fermentated states by using UPLC-QTOF-MS/MS technology, to conduct preliminary analyses, and to establish an HPLC method for the simultaneous determination of hesperidin and naringenin in Jianqu, and the variation of the two components during fermentation were compared. Methods: Waters ACQUITYTM UPLC HSST3 column (2.1 mm × 100 mm, 1.8 μm) was used; the mobile phase was 0.1% formic acid aqueous solution (A)-0.1% formic acid acetonitrile (B); The flow rate was 0.4 mL·min^−1^ with gradient elution; the column temperature was 45 °C; injection volume was 5 μL. The mass spectra of the samples were collected by negative ion mode under the electrospray ion source, and the data were screened and matched by UNIFI software. Hypersil gold C18 column (100 mm × 2.1 mm, 1.9 μm) was used; the mobile phase was acetonitrile (A)-0.1% acetic acid (B);; the flow rate with gradient elution was 0.3 mL·min^−1^; the column temperature was 30 °C; the injection volume was 2 μL. The content changes of hesperetin and naringenin in Jianqu at different fermentation time were detected. Results: A total of 54 compounds were identified, including flavonoids, amino acids, organic acids, terpenoids, coumarins, lignans, and other compounds. Under the selected HPLC conditions, the linear relationship between hesperidin and naringenin was discovered (r^2^ = 0.9996). The content of hesperidin and naringenin changed significantly in the whole fermentation process. The highest concentration of content was observed at 36 h of fermentation and then decreased to varying degrees. Conclusion: This experiment can effectively identify various chemical components in Jianqu during different fermentation periods, and determine the content of the characteristic components, so as to provide a scientific basis for further study of Jianqu fermentation processing technology as well as a sound pharmacodynamic material basis.

## 1. Introduction

Fermentation processing of traditional Chinese medicine refers to the method of foaming and undressing the medicine after purification or treatment under certain temperature and humidity conditions through the catalytic decomposition of microorganisms and enzymes [1] (p. 416). The fermentation method is different from the traditional heating and processing methods such as frying, steaming, boiling, simmering, frying, refining, etc. It has a mild effect and mainly uses microorganisms and their secreted enzymes to change the chemical composition of medicines. Due to the variety of microorganisms and the different synergistic effects among them, the same drug fermented by different methods often has different therapeutic effects. Traditional Chinese medicine fermentation converts macromolecular substances that are difficult for the human body to absorb into small molecular components easier to absorb through microbial strains, which can improve the speed of drug absorption and promote the effectiveness of drugs. Moreover, a variety of secondary metabolites are generated during fermentation, which can enhance or change the efficacy. Drug components decompose after fermentation to a certain extent, thus bringing reduced toxicity and making it relatively safe to take in. The taste of Chinese medicine is usually tough for patients, while fermentation changes the original taste into something more acceptable, and the patient’s compliance might be better. It can be seen that the fermentation processing of traditional Chinese medicine can provide new research ideas for the development of traditional Chinese medicine and has a very broad application prospect.

Jianqu, also known as Fanzhiqu and Baicaoqu, has a yellowish-brown surface, white mildew, fragrant smell, and slightly bitter taste. It has a long history of application. In addition to the effect of divine comedy, it also has the effect of deciphering the surface and reconciling it [2] (p. 385). It is produced all over the country, but Quanzhou, Fujian is the most famous, also known as Quanzhou Divine Comedy [3] (pp. 31–32). Jianqu, as a processed product fermented from dozens of fine medicinal powders, has been used by physicians in various dynasties intending to different therapeutic efficacy, and its prescriptions have changed accordingly. Jianqu is now fermented from Chinese medicines such as Artemisia annua, mint, Angelica dahurica, wheat bran, and flour. At present, there are more than 20 Jianqu products used in clinical applications. According to the literature, Jianqu can be used in the following four categories: (1) cough caused by cold. (2) cold, fever, dizziness, and vomiting caused by the plague. (3) chest stuffiness and fullness caused by qi stagnation, (4) eating and abdominal pain and indigestion caused by water and grain accumulation. At present, the research of Jianqu mainly focuses on the clinical prescription application and case analysis, and there are few reports on the chemical composition analysis of Jianqu, and the material basis of its efficacy is still unclear, which limits the clinical application of Jianqu. Therefore, clarifying the chemical composition of Jianqu before and after fermentation can provide a scientific basis for further improving the fermentation process of Jianqu and studying its pharmacodynamic material basis.

In this study, UPLC-QTOF-MS/MS technology was used to systematically analyze and study the chemical substances in Jianqu at different fermentation times, and HPLC technology was used to determine the content of the characteristic components.

## 2. Results

### 2.1. Linear Investigation Results

Taking the concentration X (mg/L) as the abscissa and the peak area Y as the ordinate, draw a standard curve to obtain the regression equations of naringenin and hesperetin. The results are shown in Table 1.

### 2.2. Methodological Investigation

#### 2.2.1. Precision Test

The test results showed that the RSDs of the peak areas of naringenin and hesperetin were 0.24% and 0.20%, respectively, indicating that the precision of the experimental instrument was good.

#### 2.2.2. Stability Test

The test results showed that the RSDs of the peak areas of naringenin and hesperetin were 0.39% and 0.33%, respectively, indicating that the test solution was stable within 12 h.

#### 2.2.3. Repeatability Test

The experimental results showed that the RSDs of the peak areas of naringenin and hesperetin were 1.95% and 1.82%, respectively, indicating that the experimental method was reproducible.

#### 2.2.4. Sample Addition Recovery Test

The test results of the sample addition recovery showed that the average recoveries of naringenin and hesperetin were 99.19% and 99.88%, and the RSDs were 2.09% and 2.20%, respectively, indicating that the experimental method was accurate. The specific results are shown in Table 2 and Table 3 below.

### 2.3. UPLC-QTOF-MS/MS Analysis Results

Comparing the basic shapes of the base peak ion chromatograms of the Jianqu samples with different fermentation times (see Figure 1), it can be intuitively found that the types of chemical components in each sample are similar. According to the response value, the chemical components in the Jianqu samples at different fermentation times were different. Data matching was conducted through UNIFI 1.8.1 software. The traditional Chinese medicine system pharmacology database and analysis platform (TCMSP) were used to search the ingredients of each single herb in Jianqu. The attribution of each component was preliminarily determined, and the component identification was carried out in combination with the reference substance and related literature information. A total of 54 compounds were identified, including 15 flavonoids, 3 amino acids, 14 organic acids, 5 terpenes, 4 coumarins, 2 lignans, and 11 other types. Among them, there are 20 species identified by the comparison of the reference substances. There are 28 unique compounds obtained through database screening. The other compounds were identified by database screening combined with literature reports. The UPLC-Q-TOF-MS/MS identification results of the compounds in Jianqu are shown in Table 4.

Jianqu mainly includes flavonoids and organic acids, Recent studies have shown that flavonoids have cough-relieving, expectorant, asthma-relieving and antibacterial activities, and at the same time have liver-protecting and detoxificating effects, and anti-free radical, antifungal, and antioxidant effects that may contribute to the treatment of hepatitis and liver cirrhosis. Organic acids have anti-inflammatory, antithrombotic, and antioxidant effects, inhibiting platelet aggregation, and inducing apoptosis of tumor cells, and the organic acids and esters in Jianqu may be the source of its wine aroma; At present, the role of these chemical components in the clinical application of Jianqu remains to be studied.

#### 2.3.1. Flavonoids

The flavonoids in Jianqu mainly come from tangerine peel, citrus aurantium, licorice, and other medicinal materials. A total of 15 flavonoids and their glycosides were identified in this experiment. Among them, 13 components were accurately identified, namely, Liquiritin, Isoquercitrin, Rutin from Pomelo Peel, Naringin, Hesperidin, Rutin, Neohesperidin, Quercetin, Luteolin, Naringenin, Hesperidin, Isorhamnetin, and Ambrosin.

Compounds 23 and 24 are the main chemical components in Jianqu and are isomers of each other. Through the data screening of the UNIFI software, combined with its accurate relative molecular mass, it can be preliminarily inferred that the two are hesperidin and neohesperidin, according to the order in which they flow out of the chromatographic column [13] (pp. 58–63). Combined with reference substances and related literature reports [4,14,15] (pp. 40–45,136–140,899–909), compound 23 was identified as hesperidin, and compound 24 was identified as neohesperidin. The same method can accurately identify compound 18 as naringin and compound 20 as naringin. Free flavonoids can exist independently in plants, and free flavonoids and corresponding carbohydrates can also be generated after hydrolysis of flavonoid glycosides. The quasi-molecular ion peaks of compounds 28 and 29 are *m/z* 271.06[M-H]^−^, *m/z* 301.07[M-H]^−^, respectively. Through the data screening of the UNIFI software, it is speculated that its molecular formulas are C_15_H_12_O_5_ and C_16_H_14_O_6_, respectively, according to its exact relative molecular mass. Combined with the relevant literature reports [4,14,15] (pp. 40–45,136–140,899–909), it can be inferred that the two are naringenin and hesperetin, and it can be identified that compound 28 is naringenin and compound 29 is hesperetin, according to the sequence of the two eluted from the chromatographic column [19] (pp. 1751–1759).

#### 2.3.2. Organic Acid Compounds

A total of 14 organic acid components were identified in this experiment, and 11 of them were identified as the follows: Quinic acid, gallic acid, caffeic acid, tartaric acid, Rosmarinic acid, Salvianolic acid C, Sparganic acid, (E,E)-9-Oxooctadeca-10,12-dienoic acid, linolenic acid, linoleic acid, and palmitic acid. The quasi-molecular ion peak of compound 52 is *m/z* 279.23[M-H]^−^, filtered through the database, so it is speculated that their molecular formulas are C_18_H_32_O_2_. Combined with accurate relative molecular mass and related literature reports [16,21,22] (pp. 20–34, 124–126, 2336–2342), it can be inferred that the compound is linoleic acid. Similarly, it can be inferred that compound 55 is palmitic acid. The quasi-molecular ion peak of compound 10 is *m/z* 179.03[M-H]^−^. After database screening, combined with the accurate relative molecular mass and the comparison of the related literature [7,8,22] (pp. 81–84, 764–774, 2336–2342), it is inferred that it is caffeic acid.

### 2.4. Assay

#### 2.4.1. HPLC Chromatogram

Take the test solution and the mixed reference solution 2, respectively, and inject them under the conditions of Section 4.3.1. The results are shown in Figure 2.

#### 2.4.2. Sample Content Determination

Precisely weigh about 1 g of S1–S4 sample, prepare the test solution according to the method under Section 4.2.2, and inject and analyze according to the analysis conditions under Section 4.3.1. The contents of naringenin and hesperetin in each sample were calculated, and the results are shown in Table 5.

## 3. Discussion

In this experiment, positive ion mode and negative ion mode were used to analyze the extract of Jianqu with different fermentation times, and the two scanning modes were compared. It was found that under the negative ion mode, the response signal of the chemical components in the sample was stronger, the peak number was more, and the mass spectrum information was clearer. Therefore, the selection is carried out in the negative ion mode.

The HPLC chromatographic binding content determination results showed that the highest content of hesperidin and naringin reached 0.2353 mg/g and 0.2821 mg/g, respectively, at 36 h of fermentation. After 36 h, both are reduced to varying degrees and may be oxidized by microorganisms. It has been reported [23] (pp. 136–142) that the optimal fermentation time of Jianqu is 30–36 h, indicating that the samples used in this experiment are qualified, and it can also be speculated that hesperidin and naringin may be the material basis for the curative effect of Jianqu. It can be clearly seen from the mass spectrum that the response values of hesperidin and neohesperidin are high. Because both are derived from dried tangerine peel and fructus aurantii, hesperidin accounts for more than 50% of the total in the dried tangerine peel [24] (pp. 36–39,52). The response value of flavonoid glycosides, such as hesperidin, neophesperidin, and naringin, gradually decreases with the continuous fermentation, and it can be inferred that their content is decreasing. Combined with the response value changes and content determination results of hesperidin and naringin, it may be because they are used as raw materials in the fermentation process to synthesize other substances or are oxidized and degraded to form other substances. At present, only the content of two index components has been detected, and the optimal fermentation processing endpoint of Jianqu cannot be comprehensively judged according to the content of these two ingredients, and follow-up research is required.

After the fermentation of Jianqu, there is a slight aroma of wine. Relevant research on wine [25] (pp. 46–50) shows that the aroma of wine is mostly the smell of acid and ester compounds. Acid esters such as linoleic acid, palmitic acid, methyl palmioleate, and ethyl linoleate were detected before and after fermentation of Jianqu liquor, which may be the source of the aroma of Jianqu liquor. Linoleic acid and linolenic acid are both unsaturated fatty acids. With the fermentation process, the response of these substances to mass spectrometry first decreased and then increased. The main reason may be that some bacteria or fungi produced at the beginning of fermentation oxidized the unsaturated fatty acids. Related research shows that [26] (pp. 82–87) the content of oleic acid and linoleic acid in some plants was significantly negatively correlated. It is speculated that enzymes may be produced later in fermentation to convert oleic acid into linoleic acid and increase its content. The response values of some small molecular organic acids, such as quinic acid and caffeic acid, decrease continuously with the progress of fermentation, and some even cannot be detected in the late fermentation, which may be because other substances are synthesized as raw materials in the fermentation process or oxidized and degraded to produce other substances.

In addition, some amino acids, such as tyrosine, begin to appear at the later stage of fermentation, which may come from the decomposition of certain substances. Glucose has been decreasing since fermentation, because as fermentation progresses, the yeast and other microorganisms produced in the fermentation system continuously consume the reducing sugar. Triterpenoids were detected 72 h later, which may be due to the decomposition of enzymes that hydrolyze triterpenoid saponins during fermentation.

In this experiment, the chemical constituents of the different fermentation times of Jianqu were analyzed efficiently, accurately, and quickly. A total of 54 compounds were identified, including 15 flavonoids, 3 amino acids, 14 organic acids, 5 terpenes, 4 coumarins, 2 lignans, and 11 others. The main components were flavonoids and organic acids. The HPLC method was used to determine the content of characteristic components, and it was found that the contents of hesperidin and naringin were the highest after 36 h fermentation. It laid a foundation for further research on the pharmacodynamic material basis of Jianqu, the action mechanism of microorganisms in the fermentation process, and the fermentation process. After that, we will further study the role of the main components in Jianqu.

## 4. Materials and Methods

### 4.1. Materials

#### 4.1.1. Experimental Instruments

Acquity UPLC-I-Class tandem Xevo-G2-SQ-TOF mass spectrometer, Masslynx4.1 mass spectrometer workstation (Waters Corporation, Milford, MS, USA), Agilent 1260 Infinity II liquid chromatograph (Agilent Technologies Co., Ltd., Santa Clara, CA, USA), UNIFY1.7 database (Waters, Milford, MS, USA); BT25S 1/100,000 balance, BSA224S-CW 1/10,000 balance (Sartoris Technology Instrument Co., Ltd., Gottingen Germany); TG16-WS high-speed desktop centrifuge (Hunan Xiangyi Laboratory Instrument Development Co., Ltd., Changsha, China); KQ-100DE Ultrasonic Cleaner (Kunshan Ultrasonic Instrument Co., Ltd., Kunshan, China).

#### 4.1.2. Experimental Reagents

Chromatographic grade acetonitrile (batch number 19085176) and methanol (batch number 19085029) were produced by Thermofisher Company (Waltham, MA, USA), chromatography grade acetic acid (batch number 20180813) was produced by Tianjin Damao Chemical Reagent Factory (Tianjin, China), and ultrapure water was made by the laboratory. The products are shown in Table 6, all purchased from Chengdu Pusi-Biotechnology Co., Ltd., and the purity is ≥98%.

#### 4.1.3. Medicinal Materials

Wheat bran and flour were reserved in the laboratory, and the other medicinal materials used in the experiment are shown in Table 7, all purchased from Zhang Zhongjing Pharmacy(Bozhou, China).

### 4.2. Methods

#### 4.2.1. Preparation of Jianqu Samples at Different Fermentation Times

The samples were prepared according to the Jianqu preparation method contained in the “Standard Set Prescription of Traditional Chinese Medicine Preparations of drugs Book XVII (Ministry of Health of the People’s Republic of China)” [2] (p. 385). In addition to wheat bran and flour, we crushed the other twenty-one drugs such as Artemisia annua into fine powder; they were mixed with wheat bran and sieved, simultaneously made into a thin paste, then mixed well with the above medicinal powder while it was still hot. The appropriate mixing ratio means it could be kneaded into balls by hand easily and dispersed upon tossing. According to method, the mixture was shaped into cubes, put in a constant temperature and humidity fermentation box with a temperature of 30 °C and a humidity of 75%, and fermented for 0, 36, 72, and 96 h, respectively. Finally, they were taken out, dried at 60 °C, and pulverized which are denoted as S1–S4, respectively.

#### 4.2.2. Preparation of the Test Solution

Samples 1 g of Jianqu were accurately weighed at different fermentation times, respectively, placed in an erlenmeyer flask, 25 mL of methanol were added, and then the total weight was weighed. The samples were sonicated for 30 min, cooled to room temperature, and methanol was added to make up for the lost weight, shaken well, and evaporated to dryness in a water bath at 70 °C. The methanol was added to the residue to dissolve and dilute to 10 mL in a volumetric flask. The solution was centrifuged at 16,149× *g* for 5 min., and the supernatant was separated and filtered through a 0.22 μm microporous membrane to obtain the test solution.

#### 4.2.3. Preparation of Reference Solution

The appropriate amount of the reference substance under item Section 4.1.2 was precisely weighed, placed in a 10 mL volumetric flask, methanol was added to dissolve and reach up to the mark, and shaken up to obtain the mixed reference substance solution 1. Appropriate amounts of hesperetin and naringenin reference substances were precisely weighed, respectively, and methanol was added to prepare the mixed reference solution 2 containing 0.5020 mg/mL and 0.5090 mg/mL, respectively.

### 4.3. Analysis Conditions

#### 4.3.1. HPLC Conditions

The column was Hypersil Gold C18 column (100 mm × 2.1 mm, 1.9 μm); the mobile phase was acetonitrile (A)-0.1% acetic acid (B); gradient elution is shown in Table 8; the flow rate was 0.3 mL·min^−1^; the column temperature was 30 °C; injection volume was 2 μL.

#### 4.3.2. UPLC-QTOF-MS/MS Analysis Conditions

First Item: Chromatographic conditions

The Chromatographic column was Waters ACQUITYTM UPLC HSST3 (2.1 mm × 100 mm, 1.8 μm); the mobile phase was 0.1% formic acid aqueous solution (A)-0.1% formic acid acetonitrile (B); the gradient elution is shown in Table 9, and the flow rate was 0.4 mL·min^−1^; the column temperature was 45 °C; injection volume was 5 μL.

Second item: Mass spectrometry conditions

Data were collected in negative ion mode under electrospray ion source, ion source temperature: 120 °C, desolvation gas: N2, flow rate: 450 L·h^−1^, temperature: 450 °C. Capillary voltage: 2000 V, cone hole voltage: 17 V, scanning range: *m/z* 100~1500. During low-energy scanning, the trap voltage was 6 eV and the transfer voltage was 4 eV. During high-energy scanning, the negative ion mode conditions were that the trap voltage was 40–60 eV, the transfer voltage was 15 eV, and the conditions of positive ion mode were that the trap voltage was 50–65 eV and the transfer voltage was 15 eV. Accurate mass LeucineenkEphalin was used as calibration solution.

### 4.4. Investigation of Linear Relationship

The mixed reference substance solution 2 under item Section 4.2.3 was accurately drawn in an appropriate amount, diluted by different times, and injected according to the analysis conditions under item Section 4.3.1. Taking the peak area as the ordinate (Y) and the concentration of the reference substance solution as the abscissa (X), we drew standard curves, respectively, to examine the linear relationship.

#### 4.4.1. Precision Test

The powder of S4 sample was accurately weighed and an appropriate amount was prepared into the test solution according to the method under item Section 4.2.2, and the test solution was continuously injected for 6 times according to the analysis conditions under item Section 4.3.1. We calculated the RSD values of the peak areas of naringenin and hesperetin, respectively.

#### 4.4.2. Stability Test

The powder of S4 sample was accurately weighed and an appropriate amount was prepared according to the method under Section 4.2.2 to prepare a solution of the test sample, which was injected at 0, 2, 4, 6, 8 and 12 h according to the analysis conditions under Section 4.3.1. We calculated the RSD values of the peak areas of naringenin and hesperetin, respectively.

#### 4.4.3. Repeatability Test

The powder of S4 sample was precisely weighed in 6 parts, prepared into the test solution according to the method under Section 4.2.2, and injected according to the analysis conditions under Section 4.3.1. We calculated the RSD values of the peak areas of naringenin and hesperetin, respectively

#### 4.4.4. Sample Addition Recovery Test

The powder of the Jianqu sample (S4) with known content was accurately weighed in 6 parts, each part was about 1 g, and an appropriate amount of mixed reference solution 2 was added, respectively, and prepared according to the method under Section 4.2.2. The test solution was injected according to the analysis conditions under Section 4.3.1. We calculated the mean recoveries and RSD values of naringenin and hesperetin, respectively.

### 4.5. Data Analysis and Processing

The database of UNIFI1.8.1 software was applied for data screening, and the intensity threshold of 3D peak detection parameters (30 counts for high energy, 200 counts for low energy) and the type of adduct ion peak (negative ion mode is [M-H]^−^, [M+HCOO] were set^−^; positive ion mode was [M+H]^+^, [M+Na]^+^). Compounds were subjected to univariate analysis using GraphPad Prism 5.01 software.

S1–S4 sample powders were prepared according to the method under Section 4.2.2 to create the test solution, and the solution were injected and analyzed according to the analysis conditions under Section 4.3.1. The peak areas of naringenin and hesperetin were measured, and the contents of each index component in the sample were calculated by regression equation.

## Figures and Tables

**Figure 1 molecules-28-00053-f001:**
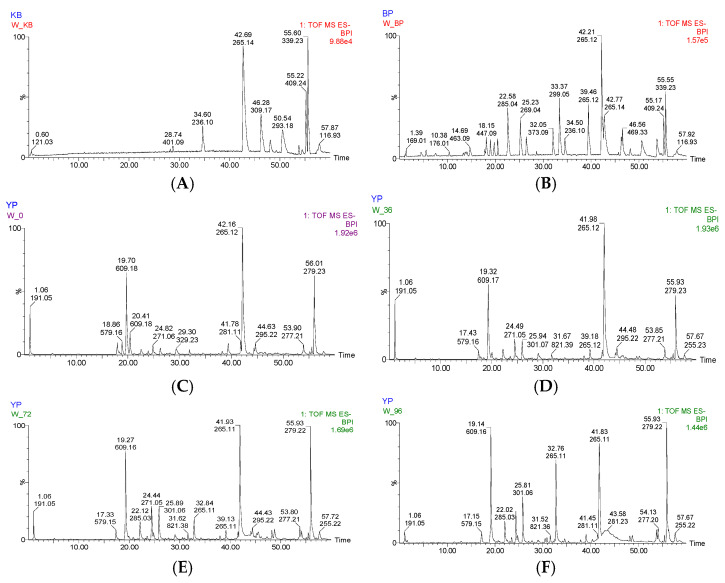
Base peak ion chromatograms of blank, standard, and Jianqu samples with different fermentation time in negative ion mode. (**A**) blank solution (**B**) reference solution (**C**) 0 h sample (**D**) 36 h sample € 72 h sample (**F**) 96 h sample.

**Figure 2 molecules-28-00053-f002:**
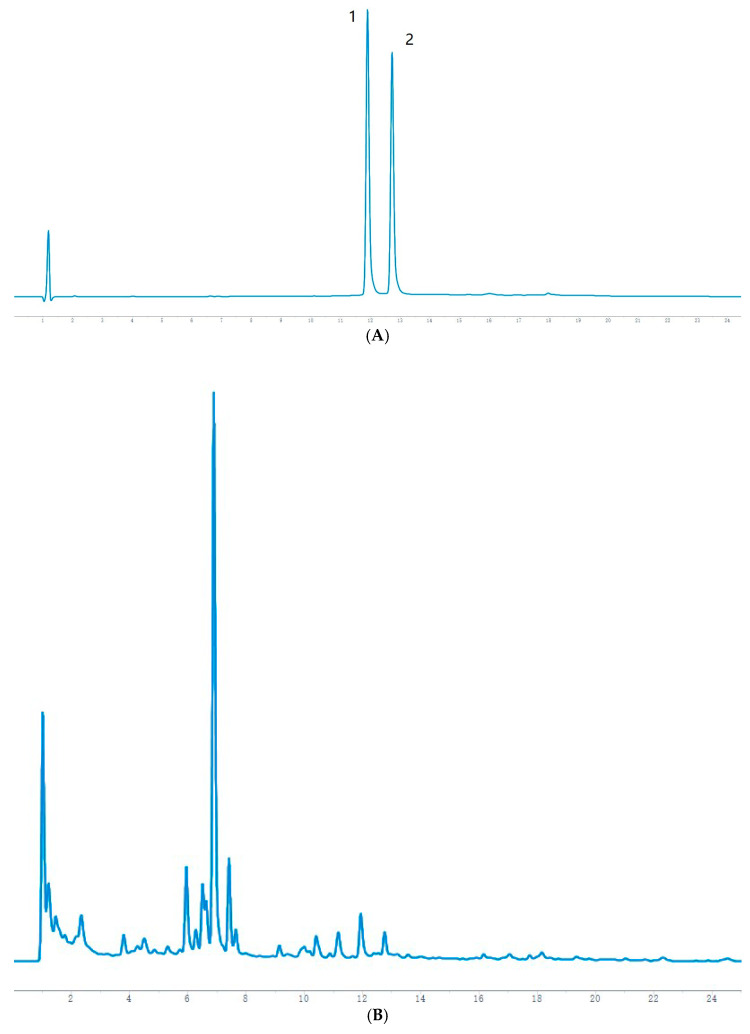
HPLC chromatograms of mixed reference substances and Jianqu samples with different fermentation times. (**A**). Mixed reference solution (**B**) 0 h sample (**C**) 36 h sample (**D**). 72 h sample (**E**) 72 h sample (**F**). 96 h sample.

**Table 1 molecules-28-00053-t001:** Results of linear investigation of naringenin and hesperetin.

Element	Regression Equation	R^2^	Linear Range (μg·mL^−1^)
Naringenin	Y = 20.679X − 34.625	0.9996	5.09–50.9
Hesperetin	Y = 17.785X + 4.7475	0.9996	5.02–50.2

**Table 2 molecules-28-00053-t002:** Test results of naringenin sample recovery rate.

Sampling Volume/mg	Sample Content/mg	Addition/mg	Measured Amount/mg	Recovery Rate/%	The Average Recovery Rate/%	RSD/%
1.0003	0.2101	0.2001	0.4109	100.35	99.19	2.09
1.0002	0.2100	0.2001	0.4100	99.95		
1.0003	0.2101	0.2001	0.4084	99.10		
1.0009	0.2102	0.2001	0.4124	101.05		
1.0006	0.2101	0.2001	0.4006	95.20		
1.0003	0.2101	0.2001	0.4101	99.50		

**Table 3 molecules-28-00053-t003:** Hesperetin sample recovery test results.

Sampling Volume/mg	Sample Content/mg	Addition/mg	Measured Amount/mg	Recovery Rate/%	The Average Recovery Rate/%	RSD/%
1.0003	0.2292	0.2036	0.4396	103.05	99.88	2.20
1.0002	0.2291	0.2036	0.4279	97.64		
1.0003	0.2292	0.2036	0.4346	99.10		
1.0009	0.2293	0.2036	0.4376	100.88		
1.0006	0.2292	0.2036	0.4276	97.45		
1.0003	0.2292	0.2036	0.4351	101.13		

**Table 4 molecules-28-00053-t004:** UPLC-Q-TOF-MS/MS identification of compounds in Jianqu.

Serial Number	tR/min	*m*/*z*[M-H]^−^	Molecular Formula	Compound	Identification Method	References
1	1.06	191.05	C_7_H_12_O_6_	quinic acid	database, literature	[4,5,6]
2	1.09	132.03	C_4_H_7_NO_4_	Aspartic acid	database ^※^	-
3	1.10	179.06	C_6_H_12_O_6_	glucose	database ^※^	-
4	1.13	149.01	C_4_H_6_O_6_	tartaric acid	database ^※^	-
5	1.17	180.07	C_9_H_11_NO_3_	tyrosine	database ^※^	-
6	1.32	169.01	C_7_H_6_O_5_	Gallic acid	database ^※^	-
7	1.32	237.04	C_10_H_8_O_4_	coumarin	database ^※^	-
8	2.66	203.08	C_11_H_12_N_2_O_2_	tryptophan	database ^※^	-
9	4.42	175.06	C_6_H_10_O_3_	3-Hydroxy-2,5-hexanedione	database ^※^	-
10	5.19	179.03	C_9_H_8_O_4_	caffeic acid	References, databases, literature	[6,7,8]
11	5.37	135.04	C_8_H_8_O_2_	Not identified	database,	-
12	6.94	340.15	C_20_H_23_NO_4_	Magnoflorine	References, databases, literature	[8]
13	7.42	785.25	C_35_H_46_O_20_	Echinacea	database ^※^	-
14	9.80	191.03	C_10_H_8_O_4_	Scopolactone	References, databases, literature	[5,6,9]
15	12.52	417.12	C_21_H_22_O_9_	liquiritin	References, databases, literature	[10,11]
16	13.02	463.09	C_21_H_20_O_12_	isoquercitrin	References, databases, literature	[12]
17	13.80	609.15	C_27_H_30_O_16_	Rutin	References, databases, literature	[4,8,12]
18	17.48	579.17	C_27_H_32_O_14_	naringin rutin	database, literature	[13,14,15]
19	18.26	445.08	C_21_H_18_O_11_	flavonoid glycosides	database ^※^	-
20	18.48	579.16	C_27_H_32_O_14_	Naringin	References, databases, literature	[4,13,14,15]
21	19.27	359.08	C_18_H_16_O_8_	rosmarinic acid	reference, database	-
22	19.63	161.02	C_9_H_6_O_3_	Umbelliferone	database ^※^	-
23	19.85	609.18	C_28_H_34_O_15_	Hesperidin	References, databases, literature	[4,13,14,15]
24	20.56	609.18	C_28_H_34_O_15_	neohesperidin	References, databases, literature	[4,13,14,15]
25	22.30	301.03	C_15_H_10_O_7_	Quercetin	References, databases, literature	[9,12,16,17]
26	22.50	285.03	C15H10O6	Luteolin	References, databases, literature	[17,18]
27	22.76	491.10	C_26_H_20_O_10_	Salvianolic acid C	database ^※^	-
28	24.82	271.06	C_15_H_12_O_5_	naringenin	References, databases, literature	[4,14,15,19]
29	26.22	301.07	C_16_H_14_O_6_	Hesperetin	References, databases, literature	[4,14,15,19]
30	26.34	315.05	C_16_H_12_O_7_	Isorhamnetin	References, databases, literature	[18]
31	29.05	219.14	C_14_H_20_O_2_	Thyme isobutyrate	database ^※^	-
32	29.30	329.23	C_18_H_34_O_5_	triangular acid	database, literature	[5,6]
33	29.80	315.05	C_16_H_12_O_7_	flavonoids	database ^※^	-
34	30.92	427.18	C_24_H_28_O_7_	coumarin	database ^※^	-
35	31.90	821.39	C_42_H_62_O_16_	Glycyrrhizinate	References, databases, literature	[10,11]
36	31.92	373.10	C_19_H_18_O_8_	cat’s eye flavin	References, databases, literature	[12]
37	36.45	487.34	C_30_H_48_O_5_	Terpenes	database ^※^	-
38	38.14	313.33	C_18_H_34_O_4_	Dibutyl Sebacate	database ^※^	-
39	39.46	265.12	C_18_H_18_O_2_	and honokiol	References, databases, literature	[8,20]
40	41.71	309.21	C_18_H_30_O_4_	9,16-Dihydroxy-10,12,14-triene-octadecanoic acid	database ^※^	-
41	41.78	281.11	C_18_H_18_O_3_	Obovatol	database ^※^	-
42	42.21	265.12	C_18_H_18_O_2_	Magnolol	References, databases, literature	[8,20]
43	44.63	295.22	C_18_H_32_O_3_	Coronaric acid	database ^※^	-
44	45.52	293.21	C_18_H_30_O_3_	(E,E)-9-Oxooctadeca-10,12-dienoic acid	database ^※^	-
45	45.57	233.15	C_15_H_22_O_2_	artemisinic acid	reference, database	-
46	48.78	295.22	C_18_H_32_O_3_	Coronaric acid	database ^※^	-
47	49.14	467.32	C_30_H_44_O_4_	Triterpenoids	database ^※^	-
48	51.89	439.25	C_27_H_36_O_5_	(25R)-Spirosta-4-ene-3,6,12-trione	database ^※^	-
49	53.90	277.21	C_18_H_30_O_2_	linolenic acid	database, literature	[16,21]
50	55.89	307.26	C_20_H_36_O_2_	Ethyl linoleate	database ^※^	
51	56.01	279.23	C_18_H_32_O_2_	Linoleic acid	database, literature	[16,21,22]
52	56.59	267.23	C_17_H_32_O_2_	methyl palmitate	database ^※^	-
53	57.05	489.36	C_30_H_50_O_5_	Triterpenoids	database ^※^	-
54	57.82	255.23	C_16_H_32_O_2_	Palmitic acid	database, literature	[5,6,16,18]

Note: “Database ※” refers to the compound that has a unique corresponding in the database screening.

**Table 5 molecules-28-00053-t005:** Determination of naringenin and hesperetin in Jianqu samples with different fermentation times (mg/g).

Sample	Naringenin	Hesperetin
S1	0.1263	0.0652
S2	0.2821	0.2353
S3	0.2746	0.2018
S4	0.2100	0.2291

**Table 6 molecules-28-00053-t006:** Experimental reference substance and batch number.

Control	Batch Number
Scutellarin	PS010076
Artemisinin	PS010373
artemisinic acid	PS020402
Gallic acid	PS000688
Costunolide	PS012275
Isorhamnetin	MUST-16120110
Luteolin	PS10320025
magnoflorine	PS012478
Quercetin	PS0605
isoquercitrin	PS010965
Pogostone	PS000401
Rutin	PS010207
Hesperidin	PS011588
neohesperidin	PS010413
Liquiritin	P29M7F12158
Glycyrrhizinate	150823
Naringin	PS012062
chrysosplenetin	PS012187
Magnolol	PS010353
Honokiol	PS011061
Atractylodin	PS011488
Scopolactone	PS011029
Nobiletin	PS012026
Kaempferol	MUST-16041502
Dehydroxylactone	PS010244
Tangeretin	PS012027
Beta-Cineol	PS010195
Oleanolic acid	PS0236-0025
Ursolic acid	MUST-15102905
rosmarinic acid	PS012101
caffeic acid	PS010522

**Table 7 molecules-28-00053-t007:** Experimental medicinal materials and batch numbers.

Medicinal Herbs	Batch Number	Manufacturer
Spicy Polygonum	20190820	Bozhou Zhang Zhongjing Chinese Herbal Pieces Co., Ltd.
cocklebur	20200103	Bozhou Zhang Zhongjing Chinese Herbal Pieces Co., Ltd.
Artemisia annua	19601CP0330	Anguo Guangming Decoction Piece Processing Factory
Bitter almonds	191101	Bozhou Zhang Zhongjing Chinese Herbal Pieces Co., Ltd.
red bean	191001	Bozhou Zhang Zhongjing Chinese Herbal Pieces Co., Ltd.
malt	190520	Henan Lvhe Pharmaceutical Co., Ltd.
Hawthorn	190801	Shandong Luan Chinese Herbal Pieces Co., Ltd.
tangerine peel	190701	Shandong Luan Chinese Herbal Pieces Co., Ltd.
Patchouli	190601	Hebei Yihe Pharmaceutical Co., Ltd.
Atractylodes	20191224	Bozhou Zhang Zhongjing Chinese Herbal Pieces Co., Ltd.
Magnolia	20191231	Bozhou Zhang Zhongjing Chinese Herbal Pieces Co., Ltd.
Muxiang	191102	Bozhou Zhang Zhongjing Chinese Herbal Pieces Co., Ltd.
Angelica dahurica	20201109	Bozhou Zhang Zhongjing Chinese Herbal Pieces Co., Ltd.
betel nut	190101CP0655	Anguo Guangming Decoction Piece Processing Factory
Citrus aurantium	20200926	Bozhou Zhang Zhongjing Chinese Herbal Pieces Co., Ltd.
Basil leaves	20190801406	Hebei Xiuhe Pharmaceutical Co., Ltd.
Mint	190601CP0680	Anguo Guangming Decoction Piece Processing Factory
Valley sprouts	20200502	Bozhou Zhang Zhongjing Chinese Herbal Pieces Co., Ltd.
Cinnamon	190801	Bozhou Zhang Zhongjing Chinese Herbal Pieces Co., Ltd.
Cypress	20191107	Bozhou Zhang Zhongjing Chinese Herbal Pieces Co., Ltd.
Licorice	20200528	Bozhou Zhang Zhongjing Chinese Herbal Pieces Co., Ltd.

**Table 8 molecules-28-00053-t008:** HPLC gradient elution method.

Time (min)	Acetonitrile (%)	0.1% Acetic Acid (%)
0→15	15→40	85→60
15→20	40→15	60→85
20→25	15	85

**Table 9 molecules-28-00053-t009:** UPLC-QTOF-MS/MS gradient elution mode.

Time (min)	0.1% Formic Acid Aqueous Solution (%)	0.1% Acetonitrile Formate (%)
0→12	95→88	5→12
12→23	88→75	12→25
23→50	75→40	25→60
50→55	40→20	60→80
55→60	20→95	80→5

## Data Availability

All the data in the manuscript are available upon reasonable request.

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
