# Peer review of "Analysis of Chemical Constituents of Traditional Chinese Medicine Jianqu before and after Fermentation Based on LC-MS/MS"

_molecules, 2022, doi:10.3390/molecules28010053_

Round 1
Reviewer 1 Report
Dear authors,
I'm sorry to write this to you, but you sent wrong version of the manuscript to the Journal. This one is, apparently, much older and contains a lot of minor mistakes I'm sure you solved before you upload the manuscript: from the abstract, which lacks a lot of spaces between words (probably incompatibility between computer systems among the authors, not your fault, but you want to make your manuscript easy-to-read and nice looking) to page numbers in text references (I checked the latest issue, Molecules support normal references like [1]), methodology given as a set of "tests", including what was later moved to accuracy, precision and intra-day, inter-day reproducibility etc. Introduction is just a short fragment, not much information about Jianqu in it yet (references are available, if you do not want write it, ie. this one: 10.1016/j.jep.2020.113512), and you definitely do not want your discussion to contain this:
3. Dis3.Discussion 167 Authors should discuss the results and how they can be interpreted from the perspective of previous studies and of the working hypotheses. The findings and their implications should be discussed in the broadest context possible. Future research directions may also be highlighted.
Assignment of chromatographic peaks is not exactly happy key word either.
Thus, I will now write to the editors that this is an important topic and interesting experiment, and you will upload proper version. If you were very unlucky and re-writen the file, you want to fix it, but you are a big group, so I believe some of you will have the right copy.
I'm very sorry to do this to you, but you paper is your business card and it would look like you had the text processed by a commercial company who does not understand science above the level of coffee making, so they just applied spellchecker on it and took your money and you didn't bother to look what they gave it to you (we had such cases in Europe recently, but I do not believe this is one of them).
I'm sure this is a misunderstanding. I am really sorry again. If I am right, you will have the revision done in no time. :)
I wish your paper better luck from now on, the topic is very interesting and you did put your instrumentation to a good use.
Author Response
Dear reviewer,
Thank you for reviewing this article.I'm sorry to reply you now because of the epidemic prevention and control. I have made the following modifications to the manuscript according to your suggestions.
1.We added a space between the words between the abstract and the reference page number.
2.We added the information of Jianqu in the introduction: " Jianqu, as a processed product fermented from dozens of fine medicinal powders, has been used by physicians in various dynasties intending to different therapeutic efficacy, and its prescriptions have changed accordingly. Jianqu is now fermented from Chinese medicines such as Artemisia annua, mint, angelica dahurica, wheat bran and flour. At present, there are more than 20 Jianqu products used in clinical applications. According to literature, Jianqu can be used in the following four categories: (1) cough caused by cold. (2) cold, fever, dizziness and vomiting caused by the plague. (3) chest stuffiness and fullness caused by qi stagnation, (4) eating and abdominal pain and indigestion caused by water and grain accumulation. At present, the research of Jianqu mainly focuses on the clinical prescription application and case analysis, and there are few reports on the chemical composition analysis of Jianqu, and the material basis of its efficacy is still unclear, which limits the clinical application of Jianqu”.
- We replaced the wrong word "3. Dis3.Discussion" changed to "3.Discussion"
- The discussion explains the results of the experiment and the speculations made based on these results, adding future research directions: " After that, we will further study the role of the main components in Jianqu."
- We modified " Assignment of chromatographic peaks " to " Chromatographic peak identification ".
Kind regards,
Wang Yishuo
Reviewer 2 Report
the research framework of the article is very interesting. the authors gave effort to responds to it. the introduction is very informative with enough information to reader. the methodology is well described and can be easily repeated. the results are clearly presented but in the discussion the authors make effort to describe how they identification the compounds without pointing out their importance. hence following question can be asked:
-what is the main importance of those compounds?
- based on the obtained results did authors think about stopping fermentation at certain point which gave the highest concentration of compounds?
- what i the benefit of the fermentation?
at line 91 authors said "A total of 54 compounds were identified,Including 15 flavonoids, 3 amino acids,..." please clarify and be mor precise about number of amino acids.
Author Response
Dear reviewer,
Thank you for reviewing this article.I'm sorry to reply you now because of the epidemic prevention and control. I have made the following modifications to the manuscript according to your suggestions.
- Study of contemporary age show shave shown that flavonoids have cough-relieving, expectorant, asthma-relieving and antibacterial activities, and at the same time have liver protecting, detoxificating effects, and anti-free radical, antifungal, antioxidant effects which may contribute in hte treatment of hepatitis, liver cirrhosis, Organic acids have the anti-inflammatory, an-tithrombotic, antioxidant effects, inhibiting platelet aggregation, and inducing apoptosis of tumor cells, and organic acids and esters in Jianqu may be the source of its wine aroma; At present, the role of these chemical components in the clinical application of Jianqu remains to be studied.
- At present, only the content of two index components has been detected, and the optimal fermentation processing endpoint of Jianqu cannot be comprehensively judged according to the content of these two ingredients, and follow-up research is required.
- Traditional Chinese medicine fermentation converts macromolecular substances that are difficult for the human body to absorb into small molecular components easier to absorb through microbial strains, which can improve the speed of drug absorption and promote the effectiveness of drugs. Moreover, a variety of secondary metabolites are generated during fermentation, which can enhance or change the efficacy. Drug components decompose after fermentation to a certain extent, thus brings reduced toxicity and making it relatively safe to take in. The taste of Chinese medicine is usually tough for patients, while fermentation changes the original taste into more acceptable, and the patient's compliance might be better.
- The amino acids identified by examination are aspartic acid, tyrosine, tryptophan 3 kinds.
Kind regards,
Wang Yishuo

Round 2
Reviewer 2 Report
the changes that author made improved the paper and now it's easier to read and understand the experiments and presented results.